# Click-electrochemistry for the rapid labeling of virus, bacteria and cell surfaces

Sébastien Depienne[1], Mohammed Bouzelha[2], Emmanuelle Courtois[3], Karine Pavageau[2], Pierre-Alban Lalys[1], Maia Marchand[1], Dimitri Alvarez-Dorta [1,4], Steven Nedellec[5], Laura Marín-Fernández[3], Cyrille Grandjean [3], Mohammed Boujtita [1], David Deniaud [1], Mathieu Mével[2] ✉ & Sébastien G. Gouin [1] ✉

Methods for direct covalent ligation of microorganism surfaces remain poorly reported, and mostly based on metabolic engineering for bacteria and cells functionalization. While effective, a faster method avoiding the bio-incorporation step would be highly complementary. Here, we used *N*-methylluminol (NML), a fully tyrosine-selective protein anchoring group after one-electron oxidation, to label the surface of viruses, living bacteria and cells. The functionalization was performed electrochemically and in situ by applying an electric potential to aqueous buffered solutions of tagged NML containing the viruses, bacteria or cells. The broad applicability of the click-electrochemistry method was explored on recombinant adeno-associated viruses (rAAV2), *Escherichia coli* (Gram-) and *Staphyloccocus epidermidis* (Gram + ) bacterial strains, and HEK293 and HeLa eukaryotic cell lines. Surface electro-conjugation was achieved in minutes to yield functionalized rAAV2 that conserved both structural integrity and infectivity properties, and living bacteria and cell lines that were still alive and able to divide.

Cell surface-associated proteins play a pivotal role in many biological events including intercellular communications and binding, signal transduction, and host-pathogen interactions. A deeper understanding of these biological processes relies on our ability to pattern cell surfaces with specific probes, drugs, binding proteins or carbohydrates. Genetic or chemical modification of viral capsids, and bacterial and cell membranes has already demonstrated interesting outlooks for a wide range of therapeutic approaches. In the field of gene therapy, genetically or chemically engineered viral capsids with peptides or carbohydrates may improve gene delivery in specific organs and target a wide variety of cell types including cardiomyoblasts, liver, tumor, smooth muscle and endothelium[1–3]. The modification of bacterial cell surfaces has also been extensively studied for diagnostic and therapeutic purposes. Research studies using living bacteria to treat solid tumors has seen considerable progress in recent years[4]. Conventional chemotherapeutics show low penetration and limited accumulation in the poorly vascularized hypoxic areas of tumors. However, anaerobic living bacteria easily colonize and continue to proliferate in these necrotic and deep regions. There, they can initiate antitumor immune response, meaning that they represent complementary avenues for cancer treatment[5]. Interestingly, this bacterial-based anticancer therapy may be significantly improved by the chemical modification of the bacterial surface with immune checkpoint inhibitors, chemotherapeutic drugs, tumor-specific antigens and photothermal sensitizers[6–8]. Last but not least, the chemical reprogramming of eukaryotic cell surfaces is a rapidly evolving research field with strong potential for drug delivery, cell-based therapies and tissue engineering[9,10]. Cell surface remodeling may be

[1]Nantes Université, CNRS, CEISAM UMR 6230, F-44000 Nantes, France. [2]Nantes Université, TaRGeT, Translational Research for Gene Therapies, CHU Nantes, INSERM, UMR 1089, F-44000 Nantes, France. [3]Nantes Université, CNRS, US2B, UMR 6286, F-44000 Nantes, France. [4]Capacités, 16 rue des marchandises, 44200 Nantes, France. [5]Nantes Université, CHU Nantes, CNRS, Inserm, BioCore, US16, SFR Bonamy, Nantes, France. ✉e-mail: mathieu.mevel@univ-nantes.fr; sebastien.gouin@univ-nantes.fr

used to: (i) improve targeting, as illustrated on mesenchymal stem cells carrying a modified sugar coat with E-selectin ligands for enhanced bone tropism;[11] and (ii) mask selective antigens, as shown with the PEGylation of donor red blood cells in chronic transfusions to avoid immune rejections[12,13]. Thus, the modification of viral, bacterial or cell surface offers highly promising therapeutic applications for a wide range of diseases.

The conjugation of proteins may be achieved with an extreme degree of site-selectivity by genetic code expansion techniques, where an unnatural amino acid (UAA) with a bioorthogonal reactive handle is site-specifically incorporated into the protein of interest by the cellular machinery[14,15]. While UAA-based protein labeling is starting to find numerous applications, this synergistic use of chemistry and biology is conducted in specialized laboratories, and there is still considerable interest for the direct modification of natural proteins. Direct chemical conjugation methods have historically been performed on the two most nucleophilic proteinogenic amino acids, i.e., the abundant lysine (K) and rare cysteine (C) residues, with various electrophilic moieties (Fig. 1a)[16]. Commercial vaccines, PEGylated drugs or antibody-drug conjugates (ADC) are mainly K and C conjugates, but modern strategies targeting less explored AA such as methionine, guanidine, tyrosine or tryptophan are being thoroughly studied in academia[17–21]. Despite the wide range of reactions now available, the direct AA modification of surface proteins from living bacteria or cells is, however, poorly represented[6,8,13,22,23]. Difficulties include the high biological complexity and chemical sensitivity of cell surfaces, cell turnover and protease degradations, or the presence of a shielding carbohydrate coat. The paradigm for chemical remodeling of bacterial and cell surfaces remains the two-step metabolic oligosaccharide engineering (MOE) method[24,25]. In MOE, a carbohydrate precursor with a biorthogonal handle is biosynthetically processed[26], then attached to cell-surface glycans for further labeling by conventional click chemistry techniques (Fig. 1b). MOE is a powerful technique and may be the best for studying cell surface glycans, but the development of direct and faster labeling methods that avoid the bacterial/cell culture step required by the biomachinery (to transform the tagged precursor) would nicely complement the toolbox of bioconjugation reactions. Such methods would particularly suit bacterial/cell lines with low propensity to incorporate synthetic sugar precursors in their glycocalyx, or organisms lacking metabolic functions, like viruses. Although electrochemical bioconjugation methods have flourished for native protein modification[27], with a recent example published by the group of Baran for chemoproteomic-based target identification[28], electrochemistry has never, to our knowledge, been applied for the exterior bioconjugation of viruses, living bacteria or cells. In this work, we describe the rapid modification of a therapeutic rAAV, living Gram −/+ bacteria and eukaryotic cells using an in situ electrochemical activation of a tyrosine-selective anchor (Fig. 1c). The electro-conjugation process occurred rapidly, yielding functionalized rAAV2 that retained their structural integrity and infectivity. Similarly, the living bacteria and cell lines remained viable and capable of division after surface modification.

## Results

A wide range of chemical methods have been developed to selectively target the tyrosine (Y) residues exposed at the surface of native

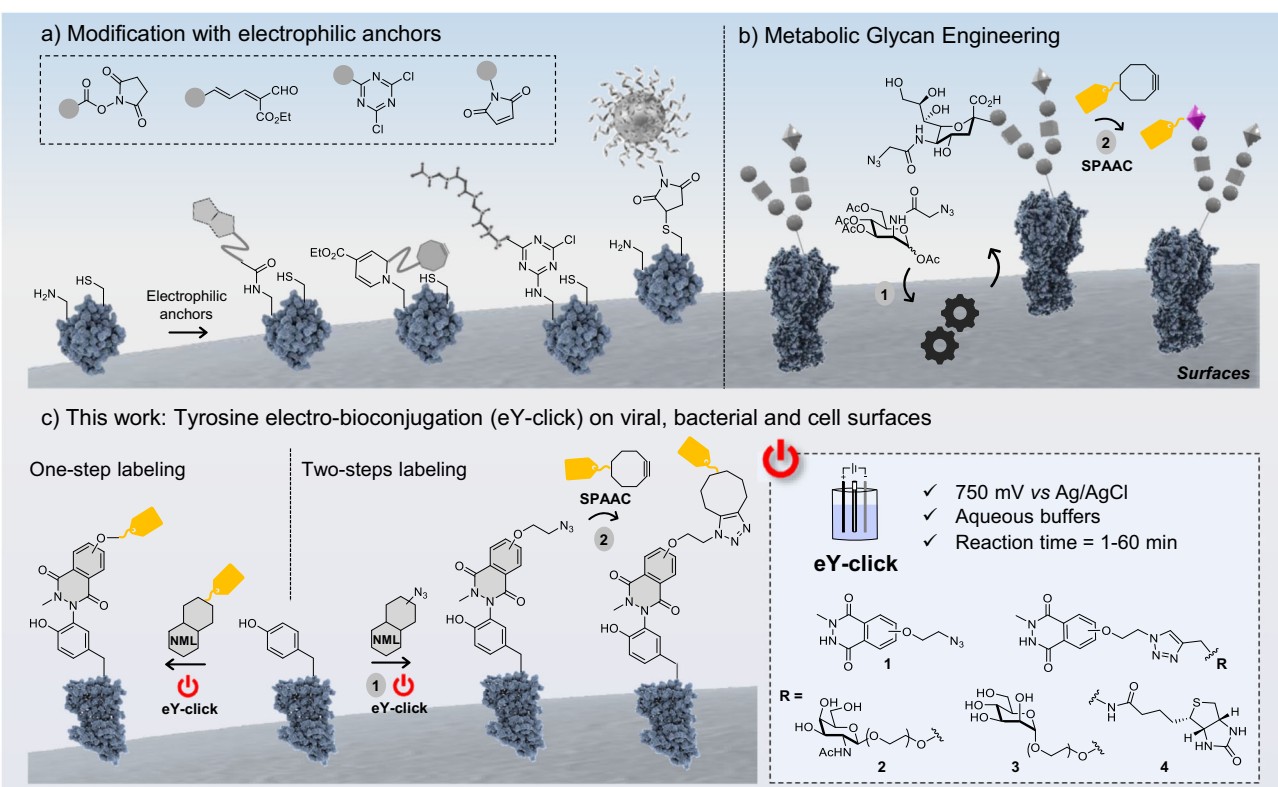

**Fig. 1 | Previously described strategies for the covalent modification of cell surfaces and present work. a** There are only a few reports on direct covalent modification of cell surfaces, with most examples restricted to electrophilic anchors targeting amino and thiol groups from lysine and cysteine. This approach is restricted to a limited number of substrates bearing non-nucleophilic functional groups. **b** MOE is a two-step labeling method where a sugar precursor (here a protected azido-tagged *N*-acetylmannosamine) is first internalized, processed by intracellular enzymes and integrated in the host glycocalyx. In a second step, the tagged sugar serves as an anchoring point for the substrate of interest thanks to a bioorthogonal reaction (here the strain-promoted azide-alkyne cycloaddition, SPAAC). **c** This work describes the first electrochemical method to modify tyrosines from viral capsids, and bacterial and cell surfaces. An electro-active anchor (luminol derivative) is activated in the presence of the (living) organisms using a 750 mV vs Ag/AgCl potential difference, applied in an electrochemical cell. This results in a rapid and soft labeling of cell surfaces by one-step or two-step approaches.

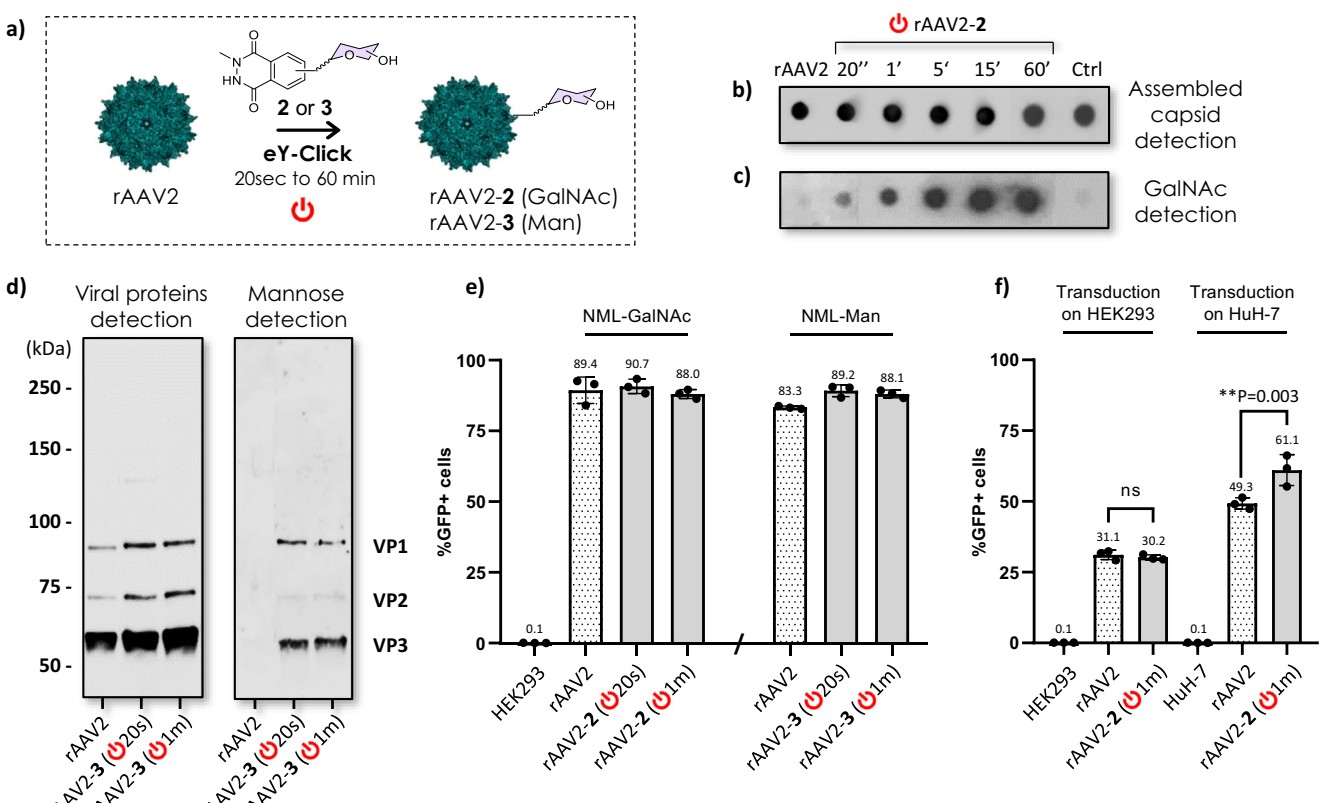

**Fig. 2 | Electrochemical coating of viral vector surface. a** rAAV2 was coated with GalNAc and Man derivatives of NML after in situ electro-activation for 20 s, and 1, 5, 15 and 60 min. **b** Dot blot analysis of viral vectors ($2.10^{10}$ vg) to detect the assembled capsid using the anti-capsid A20 antibody. **c** Dot blot analysis of viral vectors ($2.10^{10}$ vg) to detect surface-conjugated GalNAc using a labeled soybean agglutinin (GalNAc-binding lectin). Note: $4.10^{10}$ vg of rAAV2-**2**-(20 s) were deposited. **d** Western blot analysis of viral vectors ($2.10^{10}$ vg) after denaturation to detect the three constitutive viral proteins (VP1/VP2/VP3 1:1:10) using a polyclonal anti-VPs antibody (left), or to detect surface-conjugated Man using a labeled concanavalin A (Man-binding lectin). **e** %GFP+ cells quantified by flow cytometry after 48 h incubation of viral vectors (carrying a GFP reporter gene) with HEK293 cells at a MOI of $1.10^{4}$ virus/cell. **f** Comparison of transduction efficiencies of rAAV2 and rAAV2-**2** (1 min) on HEK293 and HuH-7 (expressing GalNAc receptors) cell lines at a MOI of $1.10^{3}$ virus/cell. **e**, **f** Data are shown as mean ± SD (standard deviation). Distinct samples were measured ($n = 3$). Statistical significance was assessed using one-way ANOVA tests with Dunnett's multiple comparisons (means comparison) and is presented as ns (not significant), 0.033> **$P$ > 0.002.

proteins. The groups of Barbas and Nakamura identified cyclic diazo-dicarboxyamides such as phenylurazole (PhUr)[29] or *N*-methylluminol (NML)[30,31] being among the most promising Y anchors after activation by chemical or enzymatic oxidations[32]. Inspired by these previous results, we showed that electrochemistry was a highly versatile approach for the in situ oxidative conversion of unreactive PhUr precursors into their highly Y-reactive phenyltriazolinedione equivalents[33]. The scope of this first bioconjugation method, named eY-click, was further expanded with NML derivatives showing improved kinetics after electro-oxidation (proposed reaction mechanisms Supplementary Figs. 1, 2), complete Y-chemoselectivity and the interesting possibility of double-tagging the most exposed and reactive Y, as demonstrated on peptides and proteins[34,35]. For these reasons, we selected the NML anchor in the present study.

## eY-click on recombinant adeno-associated virus rAAV2

Functionalization of free amino groups from viral capsids with carbohydrates was previously reported by our group and others to potentially improve their transduction efficiency in specific cell lines[3,36,37]. Here, we assessed the potential of electrochemistry to functionalize the Y from viral capsids using rAAV2 as a therapeutically relevant serotype. NML derivative **1** was chemically functionalized beforehand with propargyl derivatives of *N*-acetylgalactosamine (Gal-NAc) or mannose (Man) to form **2** and **3**, respectively (chemical synthesis in Supplementary Fig. 5). Sugar-coating of rAAV2 capsid was first performed with **2** in a one-step eY-click procedure for 20 s, 1, 5, 15 or 60 min (Fig. 2a). A three-electrode system (working electrode

(anode) = graphite plate, counter electrode (cathode) = platinum plate, reference electrode = Ag wire in KCl) was immersed in a neutral buffered solution of rAAV2 and **2** and an electrical potential of 750 mV vs Ag/AgCl was applied, i.e., the oxidation potential of NML in these conditions.

The electro-functionalized viruses conserved the integrity of their capsid in all cases, as observed after dot blot staining with the A20 antibody, which detects the assembled capsid (Fig. 2b). A similar assay performed with a labeled soybean agglutinin (SBA), a GalNAc-binding lectin, unambiguously confirmed the effective electro-conjugations of **2** on the surface of the viral vectors, even after 20 s voltage, with an observable time-dependent increase of staining intensity (Fig. 2c). In contrast, when the voltage was applied to a mixture of rAAV2 and a GalNAc derivative with no NML moiety (Ctrl), GalNAc could not be detected by SBA, showing the covalent anchoring of **2** to the capsid during eY-click. Interestingly, **2** labeled the rAAV2 capsid with very fast kinetics (20 s) presumably thanks to the carbohydrate moiety that drastically increases aqueous solubility of NML, improving both its diffusion rate to the working electrode and the turnover at the anodic diffusion layer for a fast and regular electron transfer. The same observations were made with Man derivative **3**, which could also efficiently label rAAV2 in 20 s eY-click. All electro-conjugated samples were subjected to capsid denaturation followed by SDS-PAGE separation on gels. The three constitutive envelope proteins (VP1/VP2/VP3 1:1:10) could be clearly identified by silver nitrate staining and western blotting using an anti-VPs polyclonal antibody. A significant mass shift of VP bands was observed for >5 min eY-click conditions for both

carbohydrates, implying a high degree of Y functionalization (Supplementary Fig. 10). Labeled lectins SBA and ConA were also used to detect GalNAc and Man, respectively, confirming the covalent conjugation of NML-carbohydrates on the three constitutive proteins of the capsid (Fig. 2d).

Next, the transduction efficiency of the electro-conjugated viral particles rAAV2-**2** and rAAV2-**3**, carrying a GFP reporter gene, was evaluated on HEK293 cell line. Cells were incubated at a multiplicity of infection (MOI) of $1.10^4$ virus/cell for 48 h and the percentage of GFP+ cells and mean fluorescence intensity (MFI) were measured by flow cytometry. Importantly, control experiment of 1 h voltage with GalNAc derivative having no luminol moiety resulted in fully conserved infectivity of the rAAV2 (Supplementary Fig. 11), proving that the applied potential difference does not alter transduction capacity. Satisfyingly, the rAAV2-**2** and rAAV2-**3** samples electro-conjugated during 20 s and 1 min fully conserved their infectivity (Fig. 2e). However, in stark contrast, longer experiments (5, 15 and 60 min eY-click) showed no GFP expression (Supplementary Fig. 11), despite the integrity of the assembled capsid being conserved. Too high a proportion of modified Y seems to impair interactions with cell surface receptors or modify the intra-cellular trafficking of the modified vectors, highlighting the crucial role of this amino acid in AAV infectivity. The level of Y modification on the rAAV2 surface should, therefore, be finely adjusted to conserve transfection and gene expression; this was reproducibly achieved here by time-controlled eY-click for 1 min. It is notable that GalNAc-decorated rAAV2-**2** (1 min) showed improved transduction capacity as compared with rAAV2, on human hepatocyte carcinoma-derived HuH-7 cells expressing GalNAc receptors (Fig. 2f). This supports the relevance of viral vector functionalization to promote transduction of specific cell lines potentially mediated by ligand-receptor interactions.

### eY-click on living bacteria

Bacteria have an inner phospholipidic cell membrane surrounded by different types of outer cell wall architectures, i.e., a thick layer of peptidoglycans and techoic acid for Gram-positive (g + ) bacteria, and a thin layer of peptidoglycans sandwiched by an exterior asymmetric bilayer of lipopolysaccharides and phospholipids for Gram-negative bacteria (g−). We investigated the capacity of eY-click to label membrane proteins from both types of cell envelopes using living *Escherichia coli* (g−) and *Staphylococcus epidermidis* (g + ) as models. The bacteria were electrochemically azido-functionalized with **1** during their exponential growth phase (optical density 0.4) in a neutral PBS solution (pH 7.4), followed by fluorescent detection using strain-promoted azide-alkyne cycloaddition (SPAAC) with the rhodamine probe DBCO-PEG$_4$-CR110 **5a** (Fig. 3a). The azido-armed bacteria were washed by repeated centrifugation and rinsing sequences, and SPAAC labeling was performed with **5a** for 1 h, followed by additional washing steps. This two-step sequence allows the preparation of a batch of azide-functionalized bacteria, which can then be modified by a variety of ligands or probes, in analogy with MOE.

We were pleased to see that fluorescence microscopy analysis revealed a strong visual labeling of the bacterial membranes for all *E. coli* and *S. epidermidis* samples subjected to the eY-click + SPAAC sequence (Fig. 3b). In contrast, fluorescence was barely detectable in bacteria that only experienced the incubation step with the cyclooctyne **5a**, serving as a control accounting for potential non-specific adsorption of **5a** or alkyne side reactions with cell surface thiols[38]. Mean fluorescence was also quantified by flow cytometry and average values from independent triplicates are presented in Fig. 3c. A clear time-dependent increase of fluorescence intensity was observed for both bacterial strains after electro-bioconjugation for 15, 30 and 60 min. The levels of labeling observed in <1 h of eY-click showed the efficiency of the method to label g(+) and g(−) bacteria harboring highly different cell membrane structures. Importantly, the

electrochemical process and the modification of the cell surface with **1** did not alter the viability of the bacteria as they were still able to proliferate at the same rate as unlabeled samples (Fig. 3d). Finally, we performed bacterial lysis and extracted membrane proteins (membrane fraction) from cytosolic and periplasmic proteins (cytosolic fraction) in the lysate (total proteins), followed by SDS-PAGE separation on gels. The quantity of proteins migrated in both inner and outer lanes were virtually the same as confirmed with LI-COR imaging system (Supplementary Fig. 15). Coomassie brilliant blue (CBB) staining showed a rich and diverse population of proteins in the non-fractionated (total) sample, as well as in both inner and outer fractions. To our great satisfaction, fluorescence detection exclusively revealed CR110-labeled proteins of various molecular weights in the membrane fractions, while the inner proteins and peptides were not modified (Fig. 3e).

### eY-click on living cells

Next, the two-step electro-bioconjugation method was assessed on eukaryotic cell lines with HEK293 cells derived from human embryonic kidney and HeLa cells from cervical cancer. In this study, an additional control experiment was added where **1** was incubated with the cells for 30 min without applying the 750 mV potential difference, and then subjected to incubation with DBCO-PEG$_4$-FAM probe **5b** ("OFF **1** + SPAAC"). Mean fluorescence intensity of this control was virtually identical to that of unmodified cells only incubated with **5b**. This clearly indicates that compound **1** does not tag the cell surface without electro-oxidative activation. However, when the potential difference was applied, a strong mean fluorescence signal was detected after 30 min for both cell lines (Fig. 4d: HEK293, Fig. 4e: HeLa). Next, we verified that our labeling sequence effectively occurred at the cell surface and not intracellularly. The electro-conjugated HEK293-FAM ($\lambda_{ex}/\lambda_{em}$ = 490/525 nm) cells were additionally subjected to intracellular staining with DAPI ($\lambda_{ex}/\lambda_{em}$ = 350/470 nm) and membrane mapping with a labeled wheatgerm agglutinin (WGA-647, $\lambda_{ex}/\lambda_{em}$ = 650/668 nm). Individual fluorescence imaging of the probes by confocal microscopy showed unambiguous overlapping signals for WGA-647 and FAM when merged, clearly standing for membrane functionalization during eY-click (Fig. 4b). Of note, fitness of both azido-tagged cell lines was not affected, as indicated by their conserved ability to divide in cell culture conditions (Supplementary Fig. 16) and confirmed by trypan blue (Supplementary Fig. 17), zombie yellow and annexin V (Supplementary Fig. 18) viability tests ( > 95% mean viability, Fig. 4c). Finally, we evaluated one-step functional biotinylation of HeLa cells by electro-bioconjugation of NML-biotin derivative **4** (chemical synthesis in Supplementary Fig. 8). After 30 min eY-click, surface biotinylation was probed with a labeled fluorescent streptavidin (SA-FAM) to evaluate efficiency by flow cytometry. In this assay, controls included cells only incubated with SA-FAM, and cells incubated with **4** during 30 min without electrical current followed by SA-FAM probing ("OFF **4** + SA"). We were pleased to reproducibly observe a significantly important population of fluorescent cells in the effective eY-click biotinylation experiment (Fig. 4f). As a stark contrast, almost no fluorescent cells were detected for both controls, which confirms the efficient covalent electro-labeling during eY-click.

## Discussion

The direct chemical functionalization of surface-exposed proteins on viral capsids or living bacteria and cell membranes is sparsely reported in the literature, and most approaches for bacteria and cells labeling rely on metabolic engineering methods requiring hours or days for the co-incubation step of the enzymatic transformation and surface exposure of the bioorthogonal reporter. As highlighted here, electrochemistry expands the scope of bioconjugation reactions and provides important advantages as compared with conventional chemical activations. The eY-click method does not require the

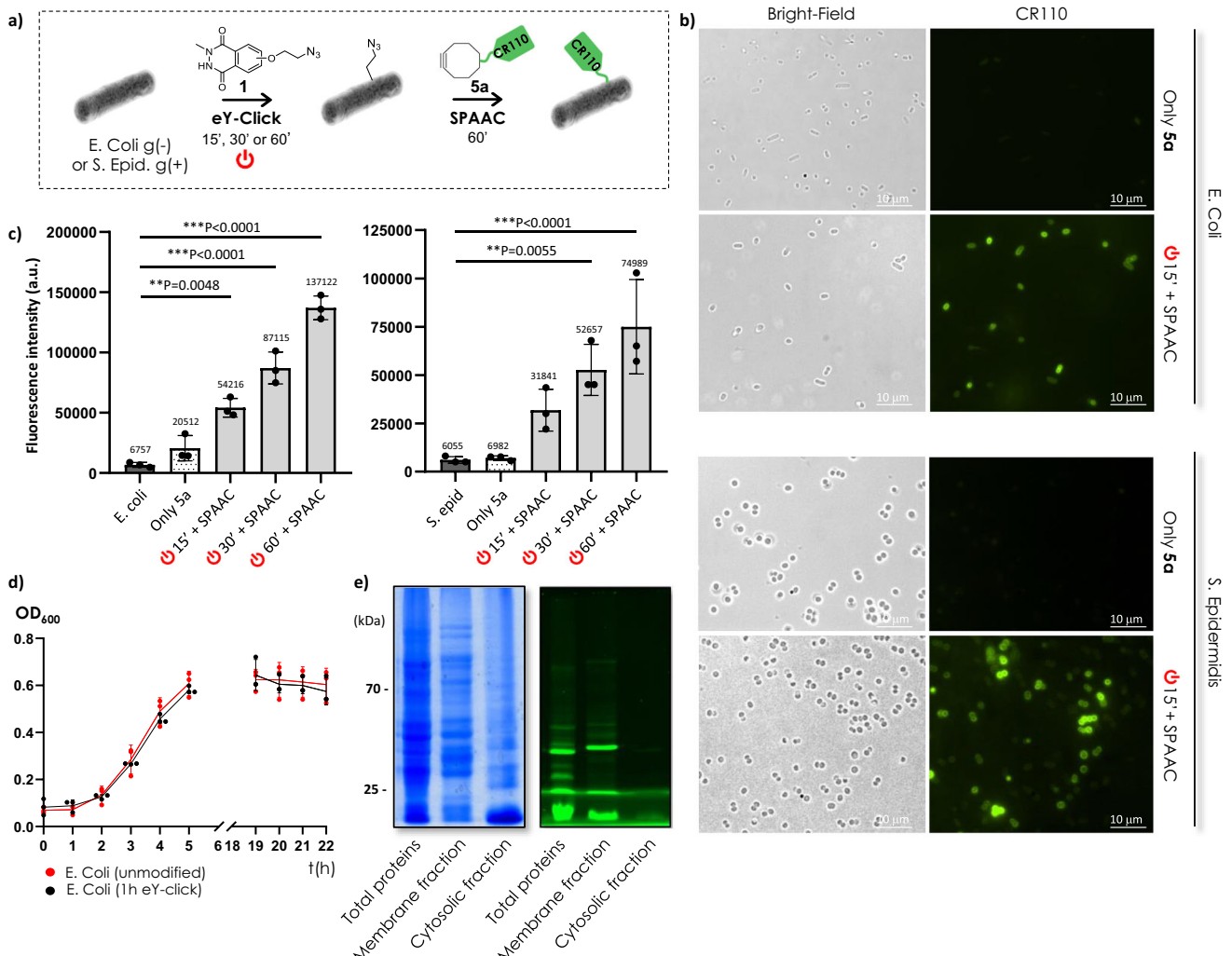

**Fig. 3 | Electrochemical coating of bacterial surfaces. a** *E. coli* g(−) and *S. epidermidis* g(+) were labeled in a two-step procedure. The bacterial strains were first electrochemically decorated with azido-luminol derivative **1** for 15, 30 or 60 min, and the covalently anchored azido-tag was reacted with the DBCO-PEG$_4$-CR110 probe **5a** by SPAAC for 1 h. **b** Optical and fluorescence images of *E. coli* and *S. epidermis* samples subjected to (i) only SPAAC (control) and (ii) eY-click and SPAAC. **c** Mean fluorescence intensity measured by flow cytometry for unmodified bacteria (*E. coli* or *S. epidermidis*), bacteria incubated with only DBCO-PEG$_4$-CR110 probe (only **5a**), and conjugation experiments by eY-click with **1** for 15, 30 or 60 min followed by SPAAC with **5a**. Data are shown as mean ± SD (standard deviation). Independent replicates were measured (*n* = 3). Statistical significance was assessed using one-way ANOVA tests with Dunnett's multiple comparisons (means comparison) and is presented as 0.033>**P > 0.002> ***P*. **d** Bacterial growth curves of **1**-labeled *E. coli* samples after 1 h eY-click compared to untreated strains. Each sample was cultured (*n* = 3) and OD$_{600}$ was regularly measured. **e** SDS-PAGE electrophoresis on gels of non-fractionated proteins (total proteins), and proteins from membrane and cytosolic fractions after bacterial lysis and fractionation protocol applied to *E. coli* strains after 1 h electro-conjugation with **1** followed by 1 h SPAAC with **5a**. Gels were colored by Coommassie Brilliant Blue (left) to confirm presence of all proteins in the fractions, and fluorescence detection at 490 nm (right) revealed fluorescent proteins in TP and MP lanes only, standing for a membrane electro-labeling.

handling of hazardous chemicals (oxidants) or potentially toxic catalysts and sensitizers for the surface modification of viruses, living bacteria or eukaryotic cells. The low electric potential applied in situ for biofunctionalization was shown to (i) preserve viral capsid architecture and its ability to transduce cells, and (ii) bacteria and cells viabilities and their capacity to divide. The click-electrochemistry showed rapid kinetics and the level of surface functionalization was reproducibly controlled in a time-dependent manner. We anticipate that click-electrochemistry will be beneficial to the investigation of complex cell surface processes and to viral, bacterial and cell-based therapies.

## Methods
The Supplementary Information provides full details of methods for the synthesis of new compounds.

## General method for electro-bioconjugation
Electro-bioconjugation experiments (eY-click) were performed with a three-electrode system using ElectraSyn 2.0 connected to SP-50 potentiostat for voltage control (see more details as supporting information). All data were recorded using EC-Lab software. For all experiments, the three-electrode system was graphite plate as anode, platinum plate as cathode, and the reference was Ag/AgCl (a thin silver rod submerged with saturated aqueous KCl solution and protected from electrolysis mixture by a porous frit glass). Before each experiment, electrodes used were thoroughly washed with EtOH (SDS for AAV experiments) and distilled water, and working electrode was re-polished on high grit sand paper (<1200 grit) to prevent potential passivation. Then, to a solution of biological target and appropriate NML derivative, 750 mV vs Ag/AgCl were applied during the studied time at room temperature. Depending on the biological target,

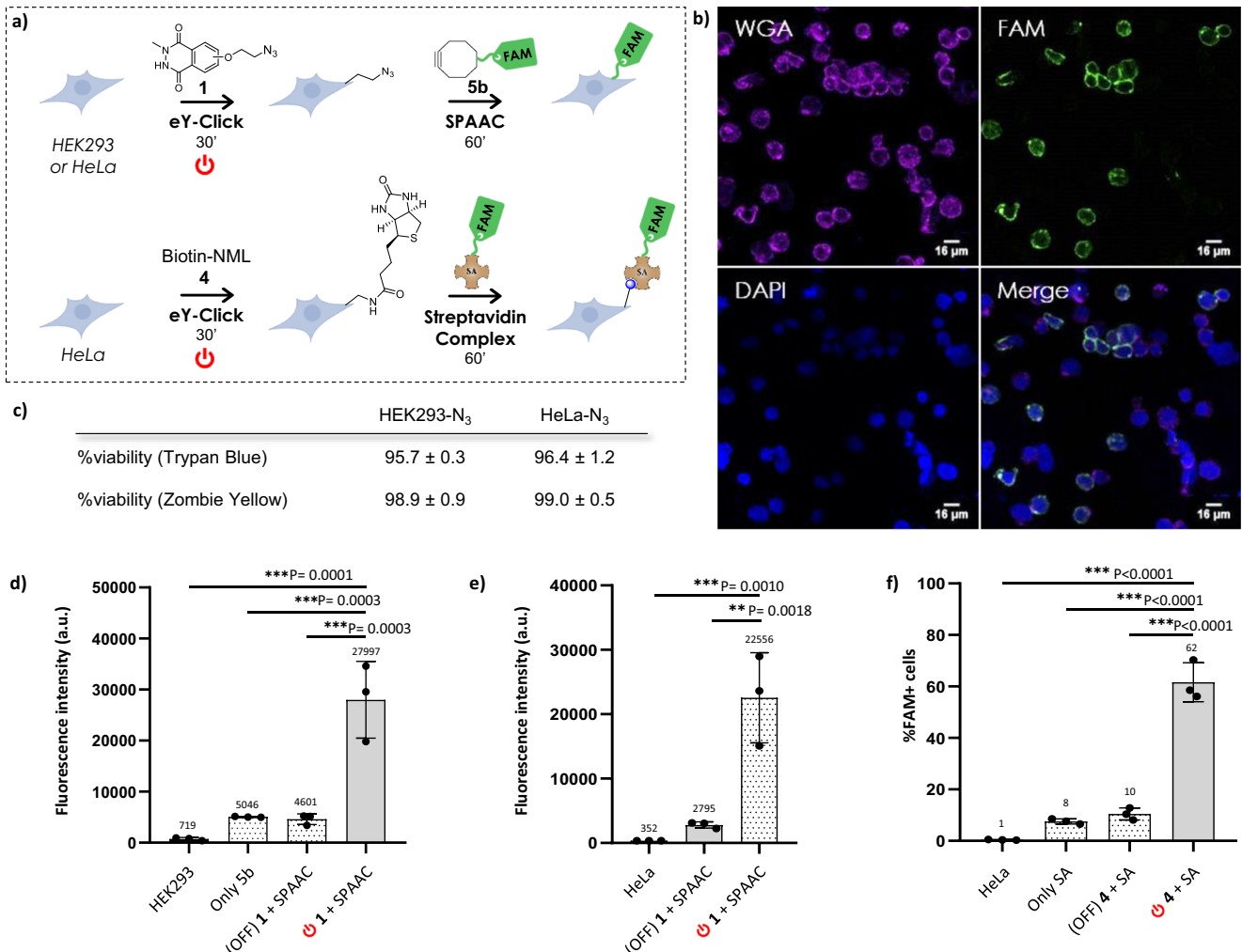

**Fig. 4 | Electrochemical coating of eukaryotic cell surfaces. a** HEK293 and HeLa cells were conjugated by 30 min eY-click with **1** or **4**, followed by 1 h SPAAC or streptavidine fluorescent (FAM) probing respectively. **b** Confocal microscopy images of HEK293 cells treated by 30 min eY-click with **1** and 1 h SPAAC with **5b**. WGA stands for membrane staining, DAPI is nuclei staining, FAM is the fluorescent probe **5b**. Merge image showcases co-localization of WGA and FAM. Scale are given bottom right. **c** Viability percentage of electro-conjugated samples as evaluated by both automatized trypan blue and zombie yellow assays (*n* = 3). **d** Mean fluorescence intensity measured by flow cytometry for unmodified HEK293 cells, cells incubated only with DBCO-PEG₄-FAM probe **5b** for 1 h (Only **5b**), cells incubated with **1** (no electro-oxidative activation) for 30 min followed by 1 h SPAAC with **5b**, and cells

incubated with **1** under 750 mV activation for 30 min followed by 1 h SPAAC with **5b** (*n* = 3). **e** Similar analysis as (**d**), i.e., mean fluorescence intensity measured by flow cytometry with HeLa cells (*n* = 3). **f** %FAM positive cells measured by flow cytometry for unmodified HeLa cells, cells incubated only with SA-FAM for 1 h (Only SA), cells incubated with **4** (no electro-oxidative activation) for 30 min followed by 1 h probing with SA-FAM, and cells incubated with **4** under 750 mV activation for 30 min followed by 1 h probing with SA-FAM (*n* = 3). **d**–**f** Data are shown as mean ± SD (standard deviation). Independent replicates were measured (*n* = 3). Statistical significance was assessed using one-way ANOVA tests with Dunnett's multiple comparisons (means comparison) and is presented as 0.002> ***P.

experimental scale was varied between 5 mL and 1 mL with different dimensions of anode/cathode.

5 mL scale setup: Dimension of anode and cathode used was 4.0 cm × 0.8 cm × 0.15 cm (accessible surface = 7.72 cm²). Considering a 5 mL solution with 90–95% of electrodes dipping, the surface/volume ratio is estimated to be ∼1.5 cm²/mL. Electrobioconjugation experiments took place in the 5 mL ElectraSyn-adapted electrochemical setup (https://www.ika.com/fr/Produits-Lab-Eq/Kit-d%27electrochimie-csp-516/ElectraSyn-20-pro-Package-cpdt-40003261/) with electrical connections derivatized to SP-50 potentiostat with alligator clips.

1 mL scale setup. Dimension of anode and cathode used was 4.5 cm × 0.3 cm × 0.10 cm (accessible surface = 3.63 cm²). Considering a 1 mL solution with 40–50% of electrodes dipping, the surface/volume ratio is estimated to be ∼1.5 cm²/mL.

## Electro-bioconjugation on rAAV2

In a 2 mL low-binding vial, 100 μL of AAV2-GFP (1ᴱ12 vg in dPBS pH 7.4, produced by the vector production center-CPV of Nantes UMR1089 gene therapy laboratory) were added to 900 μL dPBS pH 7.4 containing the appropriate quantity of carbohydrate luminol derivative **2** or **3** (final conc. 0.5 mM or 5 mM). The 1 mL scale setup was assembled and 750 mV vs Ag/AgCl were applied during the studied time at room temperature under gentle orbital shaking. After modification, the excess of unreacted luminol anchor was removed by dialysis in four successive rounds against dPBS pH 7.4 ( + 0.001% Poloxamer) in a 10 kDa MWCO cassette.

Titration of viral genomes (vg): To determine the titer (vg/mL) of all AAV samples, 3 μL were treated with 20 units of DNase I (Roche #04716728001) at 37 °C for 45 min to remove residual DNA in vector samples. After treatment with DNase I, 20 μL of proteinase K

(20 mg/mL, from MACHEREY-NAGEL®) was added and the mixture was incubated at 70 °C for 20 min. An extraction column (NucleoSpin®RNA Virus) was then used to extract DNA from purified AAV vectors. Quantitative real-time PCR (qPCR) was performed with a StepOne-Plus™ Real-Time PCR System Upgrade (Life Technologies). All PCRs were performed with a final volume of 20 μL, including primers and probes targeting the ITR2 sequence, PCR Master Mix (TaKaRa), and 5 μL of template DNA (plasmid standard or sample DNA)[3]. qPCR was carried out with an initial denaturation step at 95 °C for 20 s, followed by 45 cycles of denaturation at 95 °C for 1 s and annealing/extension at 56 °C for 20 s. Plasmid standards were generated with seven serial dilutions (containing 108 to 102 plasmid copies).

For dot blot analysis, nitrocellulose membrane was soaked briefly in PBS prior to assembling the dot blot manifold (BioRad), then AAV vectors ($2.10^{10}$ vg) were loaded. The obtained nitrocellulose membrane was then treated for the appropriate characterization (see below: capsid integrity or carbohydrate detection).

For silver nitrate or western blot, all AAV vectors ($2.10^{10}$ vg) were denatured at 100 °C for 5 min using Laemmli sample buffer (5 μL) and separated by SDS-PAGE on 10% Tris-glycine polyacrylamide gels (Life Technologies). Precision Plus Protein All Blue Standards (BioRad) were used as a molecular-weight size marker. After electrophoresis at 120 V during 200 min, gels were either silver stained (PlusOne Silver Staining Kit, Protein from GE Healthcare®) or transferred onto nitrocellulose membranes for Western blot. A 25 mM Tris/192 mM glycine/0.1 (w/v) SDS/20% MeOH buffer was used to transfer proteins during 10 min at 150 mA in a Trans-Blot SD Semi-Dry Transfer Cell (from BioRad®). The obtained nitrocellulose membrane was then treated for the appropriate characterization (See below: viral capsid proteins or carbohydrate detection).

Capsid integrity: Membrane was saturated for 2 h at RT with PBS containing 5% semi-skimmed milk and 0.1% tween. After saturation, the membrane was probed with primary antibody to mouse anti-capsid A20 (from Kleinschmidt®, diluted in milk solution 1:20) overnight at 4 °C. Then, membrane was washed thrice 15 min at RT with PBS-0.1%Tween, and probed with secondary antibody anti-mouse-HRP (from Dako®, diluted in milk solution 1:2000) during 1h30 at RT. Membrane was finally washed thrice 15 min at RT with PBS-0.1%Tween and detection of bands was performed by local treatment with $H_2O_2$/luminol during 1 min followed by chemiluminescence visualization on X-ray films.

Viral capsid proteins detection: Membrane was saturated for 2 h at RT with PBS containing 5% semi-skimmed milk and 0.1% tween. After saturation, the membrane was probed with primary antibody to rabbit polyclonal anti-AAV capsid proteins (from PROGEN Biotechnik®, diluted in milk solution 1:2000) overnight at 4 °C. Then, membrane was washed thrice 15 min at RT with PBS-0.1%Tween, and probed with secondary antibody anti-rabbit-HRP (from Jackson®, diluted in milk solution 1:20000) during 1h30 at RT. Membrane was finally washed thrice 15 min at RT with PBS-0.1%Tween and detection of bands was performed by local treatment with $H_2O_2$/luminol during 1 min followed by chemiluminescence visualization on X-ray films.

Carbohydrate detection: Membrane was saturated for 2 h at RT with PBS containing 1% gelatin, 0.1% igepal and 0.1% tween. After saturation, the membrane was probed with Soybean Agglutinin-Fluorescein lectin (from Vector Laboratories®, diluted in PBS-0.1% Tween solution 1:200) for GalNAc detection or Concanavalin A-Fluorescein lectin (from Vector Laboratories®, diluted in PBS-0.1% Tween solution 1:200) for mannose detection, overnight at 4 °C. Then, membrane was washed thrice 15 min at RT with PBS-0.1%Tween, and probed with secondary antibody anti-Fluorescein-HRP (from abcam®, diluted in PBS-0.1%Tween solution 1:5000) during 1h30 at RT. Membrane was finally washed thrice 15 min at RT with PBS-0.1%Tween and detection of bands was performed by local treatment with $H_2O_2$/luminol during 1 min followed by chemiluminescence visualization on X-ray films.

In vitro transduction: The infectivity of each sample was measured as follows. HUH7 or HEK293 cells were seeded in DMEM with 10% FBS serum and 1% penicillin-streptomycin in 6-well culture plates at a density of $10^6$ cells/well. Cells were then incubated overnight at 37 °C with 5% $CO_2$ to reach 50% confluence. Then, AAV samples were prepared by serial dilution considering the studied multiplicity of infection (MOI = n virus/n cells, varying from $10^3$ to $10^4$) and 2 μL of the samples were added to separate wells in the 6-well plates. The latter were incubated at 37 °C for 24 h. AAV-GFP-infected cells were detected and quantified by fluorescence microscopy and flow cytometry on a BD-LSRII Flow Cytometer (BD Bioscience). All data were processed by FlowJo (V10, Flowjo LLC, Ashland, OR).

**Electro-bioconjugation on bacterial strains**

TOP10 *Escherichia coli* or *Staphylococcus epidermidis* strains were precultured in lysogeny broth (LB) until optical density (OD) 0.4 approx. Then, 250 μL of preculture media were diluted in 25 mL LB and incubated at 450 g and 20 °C during 16 h (OD = 0.4 approx.). Bacteria were centrifuged at 20,000 g during 5 min, supernatant was withdrawn followed by resuspension in 25 mL PBS pH 7.4. Next, to 2.5 mL of the bacterial strain (*E. coli* or *S. epidermidis*, optical density = 0.4) solution in PBS pH 7.4 were added 2.5 mL of 2 mM azide luminol derivative **1** (final conc. 1 mM) solution in PBS pH 7.4. The 5 mL scale setup was assembled and 750 mV vs Ag/AgCl were applied during the studied time at room temperature at 500 rpm. After modification, the excess of unreacted luminol anchor was removed by performing three times the following sequence: (i) centrifugation (20,000 g during 1 min), (ii) supernatant withdrawal, (iii) bacteria resuspension in 1 mL PBS. At the end of 3rd sequence, bacteria were resuspended in 190 μL PBS pH 7.4 and 10 μL of a 2 mM DBCO-PEG₄-CR110 **5a** (obtained from Jena Bioscience®) solution (final conc. 0.1 mM) in DMSO were added. The sample was incubated at 23 °C in the dark during 1 h under moderate orbital shaking. Then, the excess of unreacted cyclooctyne was removed by performing four times the previous centrifugation/removal/washings (resuspension included 0.5% DMSO for the two first sequences). At the end of 4th sequence, bacteria were resuspended in the appropriate volumes/solutions for characterizations.

Bacteria viability was evaluated by their ability to grow in culture conditions. The longest electro-conjugation conditions (1 h) was performed in triplicate and evaluated. Directly after the electro-conjugation step, 300 μL of the samples ($OD_{600}$ = 0.6) were taken and treated 2 times with the washing sequence: centrifugation (20,000 g during 2 min) - supernatant withdrawal – resuspension in 1 mL LB (final resuspension in 300 μL LB). 80 μL of the washed samples were seeded (each condition in triplicate) in a 24-well plate, diluted to 1 mL LB ($OD_{600}$ = 0.05), and incubated at 450 g at 20 °C. $OD_{600}$ were regularly measured on an Infinite M1000 Microplate reader from TECAN using Magellan Software.

Electrobioconjugation efficiency was visualized by fluorescence microscopy (FITC excitation conditions) using Nikon Eclipse NI-E microscope (data treated with NIS software) and quantified by flow cytometry using CYTOFLEX cytometer from Beckman Coulter® – Life Sciences (data treated with FlowJo software).

Proteins membrane extraction and SDS-PAGE analysis: 800 μL of each sample were split in 2 × 400 μL. The latter were centrifuged and supernatants removed. First part was resuspended in 50 μL of resuspension buffer (sodium phosphate buffer 50 mM pH 7.4/300 mM NaCl/2 mM $MgCl_2$/DNASE 1000X/Lysozyme 100X/Protease inhibitor cocktail 200X) and 10 μL of Laemmli 6X buffer were added (Total fraction). Second part was resuspended in 500 μL of resuspension buffer and lysed by 3× periodic 5 s ON/OFF ultrasonication followed by centrifugation at 12,000 g during 30 min. Supernatant was taken off and concentrated using 3 K MWCO VWR® centrifugal filters until 80 μL final volume, and 10 μL of Laemmli 6X buffer were added (Cytosolic fraction). The remaining centrifugated pellet was resuspended in 50 μL

of 8 M urea/50 mM $NaH_2PO_4$/300 mM NaCl buffer and 10 μL of Laemmli 6X buffer were added (Membrane fraction). For each sample, the 3 fractions were heated at 95 °C during 5 min (10 min for total fraction) and 10 μL (20 μL for cytosolic fractions) were deposed on a 12% acrylamide gel. Proteins were separated at 90 V during 10 min then at 150 V during 1h15. Gels were visualized under UV, washed 3 × 10 min with water and then stained with Coomassie brilliant blue: coloration overnight and 3 × 30 min discoloration with water.

### Electro-bioconjugation on cells

Electrobioconjugation procedure with azido derivative **1**: Cells (HEK293 or HeLa) were cultured with 10% FBS serum and 1% penicillin-streptomycin at 37 °C with 5% $CO_2$. The cells were trypsinized and harvested in PBS pH 7.4 at a concentration of $6.10^6$ cells/mL. Then, in a 2 mL low-binding vial, 500 μL of the cell solution (final conc. $3.10^6$ cells/mL) were added to 500 μL of a 2 mM solution of **1** in PBS pH 7.4 (final conc. 1 mM). The 1 mL scale setup was assembled and 750 mV vs Ag/AgCl were applied during the studied time at room temperature under gentle orbital shaking. After modification, the excess of **1** was removed by performing three times the following sequence: (i) centrifugation (1200 $g$ during 2 min), (ii) supernatant withdrawal, (iii) cells resuspension in 1 mL PBS. At the end of 3rd sequence, cells were resuspended in 250 μL PBS pH 7.4 and 250 μL of a 0.2 mM DBCO-$PEG_4$-FAM **5b** (from Jena Bioscience®) solution in PBS were added (final conc. 0.1 mM). The sample was incubated at 23 °C in the dark during 1 h under moderate orbital shaking. Then, the excess of unreacted cyclooctyne was removed by performing three times the previous centrifugation/removal/washings with 1 mL PBS. At the end of 3rd sequence, modified cells were resuspended in the appropriate volumes for characterizations.

Electrobioconjugation procedure with biotin derivative 4: HeLa cells were cultured with 10% FBS serum and 1% penicillin-streptomycin at 37 °C with 5% $CO_2$. The cells were trypsinized and harvested in PBS pH 7.4 at a concentration of $6.10^6$ cells/mL. Then, in a 2 mL low-binding vial, 500 μL of the cell solution (final conc. $3.10^6$ cells/mL) were added, centrifuged to remove supernatant and 1 mL of 1 mM solution of **4** in PBS pH 7.4 (final conc. 1 mM) was added (**4** has poor solubility in aqueous media and it was helped with ultra-sonic bath). The 1 mL scale setup was assembled and 750 mV vs Ag/AgCl were applied during the studied time at room temperature under gentle orbital shaking. After modification, the excess of unreacted luminol anchor was removed by performing three times the following sequence: (i) centrifugation (1200 $g$ during 2 min), (ii) supernatant withdrawal, (iii) cells resuspension in 1 mL PBS. At the end of 3rd sequence, cells were resuspended in 50 μL of a Streptavidin-FAM solution (obtained from ThermoFisher®) and incubated at 37 °C during 1 h under moderate orbital shaking. Then, non-complexed Streptavidin was removed by performing three times the previous centrifugation/removal/washings with 1 mL PBS. At the end of 3rd sequence, modified cells were resuspended in the appropriate volumes for characterizations.

Cell viability after modification was evaluated by cell culture. Unmodified, controls and conjugated cells were seeded in DMEM with 10% FBS serum and 1% penicillin-streptomycin in a 24-well culture plate and incubated at 37 °C with 5% $CO_2$. Growth ability and confluences were evaluated, quantified and compared by microscopy and using Vi-CELL XR after 24 h, 48 h and 72 h.

Viability test with trypan blue Approx. $5.10^5$ cells in 600 μL PBS were subjected to automated trypan blue viability test using Vi-CELL XR (from Beckman Coulter, Life Sciences). % of viable cells is calculated from the ratio of viable cells and total cells. Viability was also evaluated after 24 h and 48 h culture of electro-labelled cells.

Viability test with Zombie Yellow and Annexin V: Approx. $2.10^6$ electro-conjugated cells were resuspended in 1 mL of freshly prepared Zombie Yellow staining buffer (100 μL Zombie Yellow BV605 from

Biolegend® diluted in 1 mL with Brilliant Violet Stain Buffer from BD Biosciences®) and incubated at RT in the dark. Centrifugation/supernatant withdrawal/PBS washing sequence was performed twice and the cells were resuspended in 30 μL of Annexin V 1X buffer and 1.5 μL of PE-CF594 Annexin V (from Fischer Scientific®) were added. After 15 min incubation at RT in the dark, samples were diluted with 400 μL of Annexin V 1X buffer and analyzed by flow cytometry to evaluate proportions of dead cells and apoptotic cells.

Electrobioconjugation efficiency was evaluated and quantified by flow cytometry. At the end of the electro-conjugation procedure, samples were directly analyzed on BD-LSRII Flow Cytometer (BD Bioscience) considering FITC functionalization and detection.

Membrane mapping was performed by incubating samples with 100 μL of Wheat Germ Agglutinin-AF647 lectin solution (from Invitrogen™, diluted with PBS 1:1000) 30 min at 4 °C. Two centrifugation/supernatant withdrawal/washings were performed with 250 μL of Perm/Wash Buffer (BD Cytofix/Cytoperm™ Fixation/Permeabilization Solution Kit, Fisher Scientific). The cells were then permeabilized with 100 μL of Fixation/Permeabilization solution (BD Cytofix/Cytoperm™ Fixation/Permeabilization Solution Kit, Fisher Scientific) and incubated 20 min at 4 °C. Two washings were performed the same way as before and nuclei were stained with 100 μL of DAPI solution (from Sigma Aldrich®, diluted with PBS 1:1000) during 15 min at RT.

Confocal Microscopy. Fluorescence imaging of the cells was performed on a Nikon A1R confocal microscope using a 60×/1.4 objective. The different channels were recorded as follows: excitation 405 nm: emission recorded from 425 to 475 nm; excitation 488 nm: emission recorded from 500 to 550 nm; and excitation 640 nm, emission recorded from 660 to 740 nm. The gain, offset and the power of lasers were adjusted as needed. Three-dimensional digital images were collected using NIS-Elements confocal software and appropriate fluorescence filters. We acknowledge the IBISA MicroPICell facility (Biogenouest), member of the national infrastructure France-Bioimaging.

### Reporting summary

Further information on research design is available in the Nature Portfolio Reporting Summary linked to this article.

## Data availability

The data that support the findings of this study are available within the main text and Supplementary Information. Source data are provided with this paper.

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

## Acknowledgements

This work was carried out with financial support from the *Centre National de la Recherche Scientifique (CNRS)*, the *Ministère de l'Enseignement Supérieur et de la Recherche* in France and the National Agency for Research (project ECLICK ANR- 19-CE07-0021-01 for S.G. and M.Bj; ChemAAV ANR-19-CE18-0001 for M.M. and D.D.). We thank the vector core of TaRGeT, UMR 1089 (CPV, INSERM and Nantes Université, which is a bioproduction and biotherapy national integrator (ANR-22-AIBB-0001), http://umr1089.univ-nantes.fr) for the production of the rAAV vectors used in this study. We also acknowledge the IBISA MicroPICell facility (Biogenouest), member of the national infrastructure France-Bioimaging supported by the French national research agency (ANR-10-INBS-04) for confocal microscopy images.

## Author contributions

S.G. conceived the initial idea, M.M. supervised rAAV2, HEK293 and HeLa labelling. Under the guidance of S.G., M.M. and D.D.; S.D., M.M., P-A.L. and D.A-D. performed the chemical synthesis. Under the guidance of M.Bj.; S.D. optimized the eY-click protocol. S.D., M.Bz. S.N. and K.P. carried out the rAAV2, HEK293 and HeLa bioconjugations and analysis. Under the guidance of C.G.; S.D., L-M.F. and E.C. carried out the *E. coli* and *S. epidermis* bioconjugations and analysis. S.G. wrote and S.G., S.D. revised the paper.

## Competing interests

Two patent applications have been filed on the electrochemical labelling process shown in this work. Patents #: EP23305042.6. and EP23305040.0. Applicants: Nantes Université, INSERM (Institut National de la Santé Et de la Recherche Médicale), CNRS (Centre National de la Recherche Scientifique). Inventors: S.G., D.D, M.M, S.D, D. A-D., M.Bz. Status: pending. All other authors declare no competing interests.
