## [Peer review file · Nature Communications]

REVIEWER COMMENTS

Reviewer #1 (Remarks to the Author):

General comments:

The manuscript from Gouin et al reports "eY-click", a novel surface engineering strategy using tyrosine electro-bioconjugation, which can be applied on virus, bacteria and cell. The authors provide detailed data to describe the rapid reaction system based on N-methyliminol (NML) derivatives that activated by electronic oxidation and captured by nucleophilic phenol group of tyrosine. NML derivatives, such as GalNAc, Man, biotin and click handle, can labeling the surface of virus, bacteria and mammalian cell with high specificity and efficiency, showing its potential for being a programmable method. In addition, the manuscript also indicate the bio-capability of this strategy in the infection and cell-viability assay.

The paper is well written and organized, and the experiments are well performed for easily repeating, which make the paper publishable at this stage. However, regarding the biological applications and the readership interests, there are still some following questions that need to be answered.

1, There may be a mistake in Fig. 1 (a). The traditional lysine-specific reaction group, N-hydroxysuccinimide (NHS) ester, don't contain C-C unsaturated bond. The author may misuse the structure of maleimide, which react specifically with cysteine. What's more, the curved arrow in Fig. 1 is confusing, which may indicate the readers that each electrophilic reaction group attack the product. In one word, the authors need to take the modification of Fig. 1 in consideration.

2, In Fig. 2 (e), the authors quantify the %GFP+ HEK293 cells by flow cytometry after the infection of rAAV2. Although the data confirmed there is no significant change in the GFP+ percentage, the MOI of 5.103 virus/cell may be inappropriate. The excess virus will lead to high positive rate, making it easy to neglect the differences between samples. To make this conclusion more solid, the authors should show the MFI value of GFP-related parameter in flow cytometry. Another suggestion is that the authors can choose to dilute the virus to make the MOI 1.103 virus/cell, as they show in Fig. 2 (f), which can also prove the modification doesn't affect the infection ability of rAAV2.

3, In Fig. 3 (b), the authors image the modified *S. epidermis* to show the bacteria-capability of eY-click. Although the fluorescence of CR110 differs significantly after eY-click engineering, the amount of bacteria in the field of vision varies greatly. In addition, the image is identical to Fig. S12. The authors should keep the bacterial populations in both fields roughly the same, and exchange Fig. S12 to give images of other fields to make their conclusions more credible.

In summary, the current work is interesting and important, especially in the novel reaction mechanism and its potential for programmable surface engineering. But in consideration of questions mentioned above, it needs to go through some necessary modifications to meet the standard of Nature Communications.

Reviewer #2 (Remarks to the Author):

The work reports a bioconjugation tool development showcasing the use of electrochemistry for the modification of complex biological systems including cell surfaces.

One of the most significant benefits of the click-electrochemistry method is that it does not require the use of hazardous chemicals or potentially toxic catalysts and sensitizers for surface modification. This is a considerable advantage over metabolic engineering methods, which can require long co-incubation periods and can be potentially problematic.

Furthermore, the low electric potential applied in situ for biofunctionalization has been shown to preserve the structure and function of viral capsids, living bacteria, and

eukaryotic cells, allowing for continued cellular division and viability. The rapid kinetics and time-dependent control of surface functionalization are also remarkable features of this method, enabling the investigation of complex cell surface processes. Overall, the click-electrochemistry method holds significant promise for further development of viral, bacterial, and mammalian cell bioconjugation. Overall, this work represents a significant advance in the field of bioconjugation, and I enthusiastically recommend it for the publication after the following comments are addressed.

1. It would be useful if authors provided an additional Figure panel to prime the readers on the mechanism of their transformation. Reading this manuscript and their previous work (JACS 2018) it is not going to be obvious to the general reader as to what is actually going on.
2. Along these lines, it doesn't look like the authors use a separator for their bulk experiments. Does that mean that there is a possibility of Pt(II) leaching into the reaction? If so, have authors ascertained this by ICP analysis and if the latter is true, what is the mechanistic impact of this possible event on the chemistry?
3. Do authors know anything regarding the changed metabolism in their cells when these interact with electrodes. For example, are there any metal ions being effluxed at a higher rate? This is something that can routinely happen when bacterial cells are exposed to electrode-based potentials (e.g., *Joule* 4, 800–811).
4. Authors should de-abbreviate some notations. For example, WE/CE/RE (non-electrochemists will find it confusing).

Reviewer #3 (Remarks to the Author):

Novelty

This paper by Mével, Guin, and coworkers describes an electrochemical modification of protein tyrosine residues on virus, Gram(+) and (-) bacteria, and human cell surfaces. A typical traditional method for cell surface engineering utilizes metabolic labeling of cell surface glycans and click reaction. A potential advantage of this method to the traditional method is its rapid kinetics (minutes vs. days). Furthermore, the method is also useful for metabolically inactive viruses. Despite many protein modifications reported to date, few examples are applicable to living cells. Specifically, the electrochemical activation of a labeling reagent NML is a unique point. The chemistry itself was, however, previously published by the authors' and other groups (references 33 and 35).

Scientific comments

1. Figures 2d and S9. While molecular weights of VP1-3 did not change after modifications with Man-derivative 3 on the western blot (Fig. 2d), the bands moved to heavier MW for GalNAc-derivative 2 (Fig. S9). Can the authors comment on the reason for this phenomenon?
2. Fig. 2b and the corresponding description in the text (page 4), The electro-functionalized viruses conserved the integrity of their capsid in all cases, as observed after dot blot staining with the A20 antibody, which detects the assembled capsid (Fig. 2b). This description is confusing because transduction capacity was completely diminished after the 5-min electrochemical reaction.
3. Fig. 2d, Mannose detection. Can the authors rationalize the relative band intensity between VP1-3 from Tyr contents or other factors?
4. Fig 2f. The transduction enhancement by GalNAc conjugation was observed for HuH-7, but not for HEK293. This result is potentially interesting and promising for cell type-selective gene delivery. The statistical evaluation, if the difference is significant, is necessary between the right two bars---rAAV2 (49.3) and rAAV2-2 (61.1) with HuH-7.
5. After the "eY-click on living bacteria" section. The authors always used azide conjugation followed by SPAAC. Is it possible to directly conjugate fluorescent molecules covalently bound to NML?
6. Fig. 3e. It is intriguing to see only two main fluorescence bands at approximately 25

and 50 kDa were labeled, suggesting that the electrochemical modification is to some extent selective. The authors should determine the labeled proteins and discuss how the selectivity emerges.

Referee: 1

General comments: The manuscript from Gouin et al reports “eY-click”, a novel surface engineering strategy using tyrosine electro-bioconjugation, which can be applied on virus, bacteria and cell. The authors provide detailed data to describe the rapid reaction system based on N-methyl luminol (NML) derivatives that activated by electronic oxidation and captured by nucleophilic phenol group of tyrosine. NML derivatives, such as GalNAc, Man, biotin and click handle, can labeling the surface of virus, bacteria and mammalian cell with high specificity and efficiency, showing its potential for being a programmable method. In addition, the manuscript also indicates the bio-capability of this strategy in the infection and cell-viability assay. The paper is well written and organized, and the experiments are well performed for easily repeating, which make the paper publishable at this stage. However, regarding the biological applications and the readership interests, there are still some following questions that need to be answered.

1, There may be a mistake in Fig. 1 (a). The traditional lysine-specific reaction group, N-hydroxysuccinimide (NHS) ester, don't contain C-C unsaturated bond. The author may misuse the structure of maleimide, which react specifically with cysteine. What's more, the curved arrow in Fig. 1 is confusing, which may indicate the readers that each electrophilic reaction group attack the product. In one word, the authors need to take the modification of Fig. 1 in consideration.

The structure of the N-hydroxysuccinimide ester in Fig. 1a is indeed a forgotten mistake and has been corrected. We considered the remarks on Fig. 1 and proposed a revised figure for easier comprehension.

2, In Fig. 2 (e), the authors quantify the %GFP+ HEK293 cells by flow cytometry after the infection of rAAV2. Although the data confirmed there is no significant change in the GFP+ percentage, the MOI of 5.10^3 virus/cell may be inappropriate. The excess virus will lead to high positive rate, making it easy to neglect the differences between samples. To make this conclusion more solid, the authors should show the MFI value of GFP-related parameter in flow cytometry. Another suggestion is that the authors can choose to dilute the virus to make the MOI 1.10^3 virus/cell, as they show in Fig. 2 (f), which can also prove the modification doesn't affect the infection ability of rAAV2.

We thank referee 1 for the relevant remark on precise comparison between samples. The transduction experiments were performed at MOI 1.10^4 virus/cell, and not at 5.10^3 as initially stated in the paper (corrected). Please note that this specific MOI was shown to be relevant, as an on/off transduction effect was observed in electro-functionalized rAAV2 between 1- and 5-min reaction (see new Fig S11b for GalNAc). We fully agree that the MFI value of GFP parameters should be presented and Fig S11 has been implemented accordingly.

3, In Fig. 3 (b), the authors image the modified *S. epidermis* to show the bacteria-capability of eY-click. Although the fluorescence of CR110 differs significantly after eY-click engineering, the amount of bacteria in the field of vision varies greatly. In addition, the image is identical to Fig. S12. The authors should keep the bacterial populations in both fields roughly the same, and exchange Fig. S12 to give images of other fields to make their conclusions more credible.

*We considered the remarks on Fig. 3b and proposed a revised figure where a more balanced population of bacteria can be observed for *S. epidermidis* between the control “Only 5a” and the eY-click experiment “15' ON + SPAAC”. In addition, we included different images of other fields in the supporting information (Fig S13) to support the result.*

In summary, the current work is interesting and important, especially in the novel reaction mechanism and its potential for programmable surface engineering. But in consideration of questions mentioned above, it needs to go through some necessary modifications to meet the standard of Nature Communications.

Referee: 2

The work reports a bioconjugation tool development showcasing the use of electrochemistry for the modification of complex biological systems including cell surfaces. One of the most significant benefits of the click-electrochemistry method is that it does not require the use of hazardous chemicals or potentially toxic catalysts and sensitizers for surface modification. This is a considerable advantage over metabolic engineering methods, which can require long co-incubation periods and can be potentially problematic. Furthermore, the low electric potential applied in situ for biofunctionalization has been shown to preserve the structure and function of viral capsids, living bacteria, and eukaryotic cells, allowing for continued cellular division and viability. The rapid kinetics and time-dependent control of surface functionalization are also remarkable features of this method, enabling the investigation of complex cell surface processes. Overall, the click-electrochemistry method holds significant promise for further development of viral, bacterial, and mammalian cell bioconjugation. Overall, this work represents a significant advance in the field of bioconjugation, and I enthusiastically recommend it for the publication after the following comments are addressed.

1. It would be useful if authors provided an additional Figure panel to prime the readers on the mechanism of their transformation. Reading this manuscript and their previous work (JACS 2018) it is not going to be obvious to the general reader as to what is actually going on.

We agree and a new figure (S1) is now presented in the supplementary information with a discussion on the reaction mechanism.

2. Along these lines, it doesn't look like the authors use a separator for their bulk experiments. Does that mean that there is a possibility of Pt(II) leaching into the reaction? If so, have authors ascertained this by ICP analysis and if the latter is true, what is the mechanistic impact of this possible event on the chemistry?

Experiments were indeed performed in an undivided electrochemical cell (no separation between anode and cathode), for experimental simplicity of the reaction. However, it doesn't mean that Pt²⁺ can be leached into the reaction because platinum electrode is used as a counter cathode of an oxidative event, and not as a counter anode of a reduction event. In more details, the reaction is based on the electro-oxidation of NML at the anode, thus the platinum cathode is the center of a reduction event (most likely partial reduction of oxidized NML) to counter-balance the electro-oxidation happening at the anode. Thus, platinum electrode receives electrons from the system and cannot be oxidized to leach Pt²⁺.

Partial leaching of [M]²⁺ can happen when metallic electrodes such as platinum or magnesium are used as anodic counter-electrodes, in electro-reduction reactions. In these cases, the anode is forced to give electrons to the solution and portions of the metallic electrode itself can indeed be leached in the

solution (sacrificial anode). Here, this is not the case because the eY-click reaction is based on an electro-oxidation (of NML specie) and platinum is the cathodic counter-electrode.

3. Do authors know anything regarding the changed metabolism in their cells when these interact with electrodes. For example, are there any metal ions being effluxed at a higher rate? This is something that can routinely happen when bacterial cells are exposed to electrode-based potentials (e.g., Joule 4, 800–811).

As shown in the paper, bacteria and cells survived and fully conserved their ability to divide after the electrochemical process. It is however possible that a subtle change in the metabolism may occur during the short electrochemical process. We also suppose that cells are minimally exposed to the electrodes because of their lower diffusion coefficient, especially considering that there is no magnetic stirring in the solution (only a moderate orbital shaking).

4. Authors should de-abbreviate some notations. For example, WE/CE/RE (non-electrochemists will find it confusing).

We thank Referee 2 for this remark. The roles of electrodes were de-abbreviated in the article for easier general understanding.

Referee: 3

Novelty: This paper by Mével, Gouin, and coworkers describes an electrochemical modification of protein tyrosine residues on virus, Gram(+) and (-) bacteria, and human cell surfaces. A typical traditional method for cell surface engineering utilizes metabolic labeling of cell surface glycans and click reaction. A potential advantage of this method to the traditional method is its rapid kinetics (minutes vs. days). Furthermore, the method is also useful for metabolically inactive viruses. Despite many protein modifications reported to date, few examples are applicable to living cells. Specifically, the electrochemical activation of a labeling reagent NML is a unique point. The chemistry itself was, however, previously published by the authors' and other groups (references 33 and 35).

Scientific comments

1. Figures 2d and S9. While molecular weights of VP1-3 did not change after modifications with Man-derivative 3 on the western blot (Fig. 2d), the bands moved to heavier MW for GalNAc-derivative 2 (Fig. S9). Can the authors comment on the reason for this phenomenon?

Please note that the comparison should be made at exact same duration of electro-coupling, as the level of Y-functionalization (and increase in protein molecular weight) is time-dependant. If so, experiments and analysis with both glycosides derivatives 2 and 3 are consistent and the same trends are observed. A rearranged figure is proposed in the supporting information (Fig. S10, implemented with rAAV2-2-20s) for a clearer overall observation and understanding.

2. Fig. 2b and the corresponding description in the text (page 4), The electro-functionalized viruses conserved the integrity of their capsid in all cases, as observed after dot blot staining with the A20 antibody, which detects the assembled capsid (Fig. 2b). This description is confusing because transduction capacity was completely diminished after the 5-min electrochemical reaction.

An intact assembled capsid isn't the only decisive parameter to allow the viral vector to achieve every steps of the complex pathways of transduction (from cell internalisation to gene takeover and GFP expression). Nature and position of some amino acids of the VPs indeed play pivotal roles in the transduction events (e.g. interactions with cell-surface receptors, intracellular trafficking, protein-mediated endosomal escape, etc).

Here, electro-bioconjugation of rAAV2 capsid with NML-glycoside derivatives did not compromise the assembled capsid integrity as confirmed with A20 antibody (Fig 2b for GalNAc and Fig S10a for mannose), and this was the case for all the experiments (20 sec to 60 min). Time-controlled electro-bioconjugations resulted in tuned degree of Y-functionalization (from low to high) at the surface of the assembled capsids (as observed with Dot Blot and Western Blot analysis using GalNAc/Man detection protocols, and with the mass shifts of VPs bands). Hence, we suppose that during >5 min experiments some determinant tyrosines were conjugated and could then not be involved in some of the transduction events that they standardly regulate.

We thank Referee 3 for this remark and a short comment was included in the main text to clarify that infectivity was lost despite the structural integrity of vectors being conserved.

3. Fig. 2d, Mannose detection. Can the authors rationalize the relative band intensity between VP1-3 from Tyr contents or other factors?

Capsid of the rAAV2 vector is composed of three proteins: VP1 (~90 kDa), VP2 (~75 kDa) and VP3 (~60 kDa), in a 1:1:10 ratio. One capsid is composed of 60 proteins, 5 VP1, 5 VP2 and 55 VP3. Thus, VP3 band is always much more intense than VP1 and VP2 bands in all SDS-PAGE and Western Blot analysis, which is indeed the case in Fig. 2d for mannose detection. Also, VP1 seems more visible (i.e. functionalized) than VP2. This is a trend that has also been observed by eY-click with azido derivative 1 (data not included in this manuscript). There are 26 Y on VP2 and 32 Y on VP1, that may be more accessible, which may explain this result.

4. Fig 2f. The transduction enhancement by GalNAc conjugation was observed for HuH-7, but not for HEK293. This result is potentially interesting and promising for cell type-selective gene delivery. The statistical evaluation, if the difference is significant, is necessary between the right two bars---rAAV2 (49.3) and rAAV2-2 (61.1) with HuH-7.

The statistical evaluation indeed resulted in significantly improved transduction on HuH-7 with rAAV2-2-1min as compared to rAAV2 and this was added on Fig. 2f

5. After the “eY-click on living bacteria” section. The authors always used azide conjugation followed by SPAAC. Is it possible to directly conjugate fluorescent molecules covalently bound to NML?

Fluorescein can't be directly electro-conjugated onto proteins because of the phenolic moiety (an NML-fluorescein derivative was previously synthesized and proved inefficient for electro-functionalisation of proteins). Furthermore, fluorescent molecules typically are redox actives at low potentials and the compatibility of e.g. rhodamines or cyanines with the NML-eY-click reaction needs to be studied in a case-by-case manner to identify if that can be considered. Thus, intensive investigations remain to be done to identify compatible fluorochromes, and in this study we instead used a convergent SPAAC-based approach for bacteria and cells fluorescent-labelling.

6. Fig. 3e. It is intriguing to see only two main fluorescence bands at approximately 25 and 50 kDa were labeled, suggesting that the electrochemical modification is to some extent selective. The authors should determine the labeled proteins and discuss how the selectivity emerges.

We were also intrigued to see a certain level of selectivity in the labeling of membrane proteins from E.coli. The determination of the nature of these proteins and the rationalization of the selectivity observed (nature of proteins, level of expression, number of solvent -exposed tyrosine) would certainly be of interest but this would require additional competences from other research groups. We agree that a study, beyond the scope of this manuscript, and specifically dedicated to the analysis of labelled proteins should be planned in near future.

REVIEWERS' COMMENTS

Reviewer #1 (Remarks to the Author):

This manuscript is well written and organized, and the authors provided sufficient supplementary data to answer my last two questions. My final suggestion is that in the revised Figure 1(a), the sulfydryl or amino group not involved in the reaction should be remained in product. After the authors' modifications, this article can meet the corresponding requirements of Nature Communications.

Reviewer #2 (Remarks to the Author):

Authors have addressed my comments, publication is recommended.

Reviewer #3 (Remarks to the Author):

The authors successfully addressed all the comments this reviewer raised in the initial submission. This reviewer recommends acceptance as it is.